# Prevalence of substance and hazardous alcohol use and their association with risky sexual behaviour among youth: findings from a population-based survey in Zimbabwe

Kudzai Hlahla [1,2] Steven Chifundo Azizi,[1,3] Victoria Simms [1,3]
Chido Dziva Chikwari [1,3] Ethel Dauya,[1] Tsitsi Bandason,[1]
Mandikudza Tembo [1,3] Constancia Mavodza,[1,4] Katharina Kranzer,[1,5]
Rashida Ferrand[1,5]

KH and SCA contributed equally.

For numbered affiliations see end of article.

**Correspondence to**
Dr Rashida Ferrand;
rashida.ferrand@lshtm.ac.uk

## ABSTRACT

**Objectives** Hazardous drinking (HD) and substance use (SU) can lead to disinhibited behaviour and are both growing public health problems among Southern African youths. We investigated the prevalence of SU and HD and their association with risky sexual behaviour among youth in Zimbabwe.

**Design** Data analysis from a population-based survey conducted between October 2021 and June 2022 to ascertain the outcomes of a cluster randomised trial (CHIEDZA: Trial registration number:NCT03719521). Trial Stage: Post-results.

**Setting** 24 communities in three provinces in Zimbabwe.

**Participants** Youth aged 18–24 years living in randomly selected households.

**Outcome measures** HD was defined as an Alcohol Use Disorders Identification Test score ≥8, SU was defined as ever use of ≥1 commonly used substances in the local setting.

**Results** Of 17 585 participants eligible for this analysis, 61% were women and the median age was 20 (IQR: 19–22) years. Overall, 4.5% and 7.0% of participants reported HD and SU, respectively. Men had a substantially higher prevalence than women of HD (8.2% vs 1.9%) and SU (15.1% vs 1.5%). Among men, after adjusting for socio-demographic factors, we found increased odds of having >1 sexual partner in those who engaged in SU (adjusted OR (aOR)=2.67, 95% CI: 2.21 to 3.22), HD (aOR=3.40, 95% CI: 2.71 to 4.26) and concurrent HD and SU (aOR=4.57,95% CI: 3.59 to 5.81) compared with those who did not engage in HD or SU. Similarly, there were increased odds of receiving/providing transactional sex among men who engaged in SU (aOR=2.51, 95% CI: 1.68 to 3.74), HD (aOR=3.60, 95% CI: 2.24 to 5.79), and concurrent HD and SU (aOR=7.74, 95% CI: 5.44 to 11.0). SU was associated with 22% increased odds of inconsistent condom use in men (aOR=1.22, 95% CI: 1.03 to 1.47). In women, the odds of having >1 sexual partner and having transactional sex were also increased among those who engaged in SU and HD.

**Conclusion** SU and HD are associated with sexual behaviours that increase the risk of HIV acquisition in youth. Sexual and reproductive health interventions must consider HD and SU as potential drivers of risky sexual behaviour in youths.

## INTRODUCTION

Hazardous drinking (HD) and substance use (SU) are of increasing public health concern.[1] Globally, an estimated 2.8 million deaths were attributed to alcohol use while 425 000 deaths were estimated to have been caused by SU between 1990 and 2016.[2] Alcohol is the most used substance globally while SU, defined as the use of illicit drugs such as opioids, cannabis, amphetamines and cocaine is on the increase in sub-Saharan Africa (SSA).[2–8] In 2016 Southern Africa had the second highest age-standardised burden of disease attributable to alcohol, after Eastern Europe.[2] Current SU by adolescents in many countries

has been reported to be higher than lifetime use in previous generations, with SU among adolescents and young people being predicted to increase in the African continent by 40% by 2030.[9]

Adolescence and young adulthood are periods of rapid physical, psychosocial and cognitive development due to the increasing complexity of brain development that occurs. Brain development during adolescence is characterised by slower development of the prefrontal cortex, which processes cognitive and emotional information such as prioritising, planning, rational decision-making and self-regulation.[10] In contrast, there is rapid development of the limbic system that regulates emotions, sensation seeking and determining reward and punishment.[10] This developmental disconnect in the brain is thought to result in experimentation and sensation seeking that may result in risk-taking behaviour during adolescence and young adulthood.[10] Although an essential facet of cognitive development and maturity, such experimentation and risk-taking may include the hazardous use of alcohol and substances as well as sexual behaviour that increases an individual's risk of unintended pregnancy and transmission of sexually transmitted infections (STI) such as HIV.[11 12] SU and HD impair cognitive functioning in a variety of ways including causing impaired decision-making, reduced risk perception and disinhibition.[12–14] SU and HD may alter the individual's perception of pleasure experienced from having sex under the influence of intoxicating substances, as well as physically limit an individual's ability to think of and take protective measures resulting in an increased risk of contracting HIV and other sexually transmitted infections.[1 11 14]

Studies conducted among adolescents and young adults have shown an association between high alcohol consumption and the use of illicit substances with risky sexual behaviour including inconsistent condom use and having multiple sexual partners.[15 16] However, most of these studies have been conducted in high-income countries in Europe, Asia and the USA. On conducting a literature search, we found very few studies that have been conducted, investigating HD, SU and its association with risky sexual behaviour within the African context, with the few studies that we did find being done in specific populations such as orphans, children and youth residing in the streets or in informal settlements.[1 17 18] There is a dearth of data on SU and HD among Zimbabwean youth specifically and to our knowledge, there has been no study conducted to determine the prevalence of HD and SU on a population-based level, and its association with risky sexual behaviour. The intersection between SU and HD and sexual behaviours is of particular concern in Southern Africa, which has the highest HIV prevalence of any global subregion.[19] This study therefore aimed to determine the prevalence of HD and SU and its association with sexual behaviour among youth, a population at particularly high risk of HIV and STI, in Zimbabwe.[20 21]

## METHODS

### Study design and study setting

This study used data from a population-based survey conducted to ascertain the outcome of a cluster-randomised trial (CHIEDZA) (Trial Registration number: NCT03719521). The trial protocol has been published elsewhere.[20] Briefly,[20] the CHIEDZA trial was conducted in three provinces (Harare, Bulawayo and Mashonaland East), each with eight clusters (defined as geographically demarcated areas) randomised 4:4 to either standard of care (existing facility-based health services) or the intervention which was integrated community-based HIV and sexual and reproductive health service provision.[20] The trial outcomes were ascertained through a cross-sectional population-based survey, conducted in Harare (October to December 2021), Bulawayo (January to March 2022) and Mashonaland East (April to June 2022) aiming to recruit 16800 youths, aged 18–24 years (700 per cluster).

### Survey methods

A multistage sampling method was employed in the survey: buildings within each cluster were mapped using satellite images on OpenStreetMap, and ARCGIS was used to segment streets into short sections (roughly 100–200 m long). Selected street sections were randomly sampled, and all residents within these sections were listed. Individuals aged 18–24 years living in the mapped sections were invited to participate in the survey.

An interviewer-administered questionnaire collected socio-demographic data, participants' sexual behaviour, knowledge of and use of HIV prevention methods, mental health and experience of violence. The Alcohol Use Disorders Identification Test (AUDIT) was used to screen for alcohol use disorder.[22] This is a 10-item internationally validated tool developed by the WHO and the most widely used alcohol screening instrument globally. The frequency of use and range of substances commonly used in the local setting were recorded. This questionnaire specified drugs by category, as drugs that are smoked (eg, weed, dagga, ganja), orally ingested (ganja cake or popcorn, prescription drugs such as cough syrup), sniffed or inhaled (eg, glue, cocaine) or injected (excluding medical drugs like insulin).

Survey data were collected onto encrypted and password-protected electronic tablets using SurveyCTO (Cambridge, USA) and uploaded to a secure SurveyCTO server at the end of each day. Data was downloaded and stored onto a password-controlled and secure Biomedical Research and Training Institute Microsoft SQL Server and managed using Microsoft Access as the front-end and with access limited to defined study personnel.

### Explanatory and outcome variables

An AUDIT score of 8 was used as the cut-off in accordance to WHO guidelines, with a score of 8 and above being indicative of HD and harmful alcohol use.[22] SU was defined as ever use of one or more of the substances listed above. The outcome was sexual risk behaviour. Sexual

risk behaviour was assessed using three self-reported indicators: (1) condom use during vaginal and anal sex in the past 12 months (2) having more than one sexual partner in the past 12 months, and (3) receiving or providing transactional sex as defined by receiving or providing money or help to pay for their expenses or favour in order to enter or remain in a sexual relationship in the past 12 months. The socio-demographic variables were categorised as follows: Age was divided into two groups, 18–20 years and 21–24 years. Marital status was categorised as 'never married' and 'ever married', distinguishing individuals based on their marital history. The education level was defined by three ascending categories: 'up to primary school', 'secondary school' and 'post-secondary school'. Employment status was categorised as either 'in school or formal employment' or 'informal employment or unemployed', differentiating individuals by their participation in structured employment or education. Lastly, the wealth quintile ranked individuals into five levels from 'poorest' to 'richest', based on household assets. These quintiles were calculated using principal components analysis to assign weights to household assets (fridge, bicycle, vehicle, radio, microwave, cell phone and computer or laptop or tablet). Each household was then given a factor score that placed it on a continuous scale of relative wealth. These scores were standardised to a mean of 0 and an SD of 1 and were used to divide the sample into five equal parts, known as quintiles. These quintiles represented varying levels of wealth, from the lowest (poorest) to the highest (richest).

### Data analysis

Data analysis was performed using Stata V.17 (StatCorp, Texas, USA). The prevalence of SU, HD and concurrent SU and HD among men and women was calculated. To assess the effect of HD and SU on risky sexual behaviour, a forward-stepwise multivariable regression analysis using a multilevel mixed-effects generalised linear model was used. This model was selected due to the multistage sampling with stratification and unequal sampling probabilities for the clusters because they had different numbers of youth residents.[23] The analyses were stratified by sex given the difference in prevalence of SU and HD. Data analysis was conducted using the 'meglm' command in Stata, which allows for the incorporation of weights in the analysis and uses a pseudolikelihood approach to account for the inverse probability weights. This method is known for providing robust SEs when sampling weights are present. Age group was considered an a priori confounder and the final models were adjusted for socio-demographic characteristics and trial arm as the intervention could have had an effect on sexual behaviours.

### Patient and public involvement

The research question was informed by reports by health providers, local policymakers, community

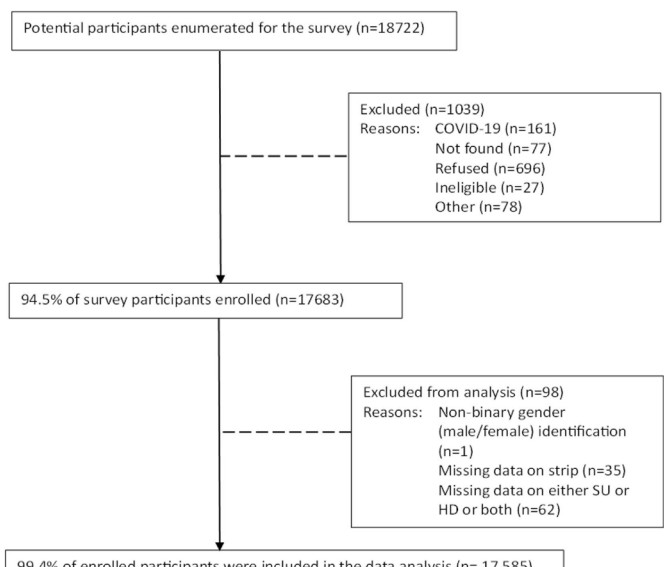

**Figure 1** Participant recruitment flow chart. HD, hazardous drinking; SU, substance use.

members and community leaders of increasing levels of SU, particularly among youth. Youth were involved in the design and delivery of the CHIEDZA intervention. The participant information video used for the informed consent process was co-designed with youth, and the questionnaire was developed and piloted with youth. Results of the CHIEDZA have been disseminated to study communities through multiple community events. A detailed mixed methods process evaluation was embedded in the CHIEDZA trial and data collection was undertaken by trained youth researchers.

## RESULTS

A total of 17 683 participants were surveyed, with 1 participant excluded from data analysis due to being the only 1 with non-binary gender identification (and therefore not possible to categorise for analyses), 35 youths excluded due to missing data and 62 with missing values on either SU or HD (see figure 1). Overall, 17 585 participants were eligible for analysis (figure 1). The median age of participants was 20 years (IQR: 19–22) and 61% were women.

### Prevalence of HD and SU

Of the 17 585 participants eligible for analysis, 15 828 (90.3%) were non-SU and non-HD. Overall, 816 participants were HD and 1254 were SU. The overall prevalence of HD was 4.5% (95% CI: 4.1% to 4.9%) and of SU was 7.0% (95% CI: 6.5% to 7.6%), with the prevalence of concurrent HD and SU being 1.8% (95% CI: 1.5% to 2.0%). Men (591) had a higher prevalence than women (225) of HD (8.2% vs 1.9%). Similarly, a higher proportion of men 1070 (15.1%) were SU compared with women 184 (1.5%). Concurrent HD and SU were reported by 262 (3.7%) men and 51

Table 1  Substance and hazardous alcohol use of male participants against socio-demographic and economic characteristics of participants, by sex

| Variable | Males | | | | |
| --- | --- | --- | --- | --- | --- |
| | | No SU or HD | SU only | HD only | HD and SU |
| | N* | n* (%)† | n* (%)† | n* (%)† | n* (%)† |
| Total | 6884 | 5485 (80.4) (95% CI: 79.1 to 81.7) | 808 (11.4) (95% CI: 10.4 to 12.4) | 329 (4.49) (95% CI: 3.93 to 5.13) | 262 (3.71) (95% CI: 3.22 to 4.28) |
| Age category in years | | | | | |
| 18–20 | 3825 | 3231 (84.8) | 381 (9.7) | 121 (3.2) | 92 (2.3) |
| 21–24 | 3059 | 2254 (74.8) | 427 (13.5) | 208 (6.2) | 170 (5.5) |
| Marital status | | | | | |
| Never married | 6528 | 5242 (81.0) | 759 (11.3) | 292 (4.2) | 235 (3.5) |
| Ever married | 356 | 243 (71.4) | 49 (11.5) | 37 (9.8) | 27 (7.3) |
| Education level | | | | | |
| Up to primary school | 248 | 190 (76.0) | 32 (12.6) | 12 (4.8) | 14 (6.6) |
| Secondary school | 6002 | 4790 (80.8) | 713 (11.4) | 274 (4.2) | 225 (3.6) |
| Post-secondary schooling | 634 | 505 (78.6) | 63 (9.9) | 43 (7.2) | 23 (4.3) |
| Wealth quintile | | | | | |
| Poorest | 1148 | 964 (85.4) | 124 (9.9) | 29 (2.5) | 31 (2.2) |
| Poorer | 1192 | 950 (81.6) | 141 (10.4) | 58 (4.5) | 43 (3.6) |
| Middle | 1443 | 1132 (79.1) | 177 (11.9) | 68 (4.5) | 66 (4.5) |
| Richer | 1462 | 1140 (77.9) | 184 (12.8) | 83 (5.5) | 55 (3.8) |
| Richest | 1635 | 1297 (78.4) | 182 (11.8) | 89 (5.3) | 67 (4.4) |
| Employment | | | | | |
| In school or formal employment | 2824 | 2386 (84.5) | 247 (8.8) | 117 (4.0) | 74 (2.7) |
| Informal employment or unemployed | 4060 | 3099 (77.8) | 561 (13.0) | 212 (4.8) | 188 (4.4) |

*Unweighted count.
†Weighted per cent.
HD, hazardous drinking ; SU, substance use.

(0.4%) women. Weed or Ganja was the most prevalent substance used in both men (14.5%) and women (1.06%), followed by swallowed drugs (1.67% in men vs 0.35% in women) and prescription drugs (1.65% in men vs 0.31% in women).

Among men, HD and SU were more common in older youth (21–24 years), ever married, in a higher wealth quintile and in those who were informally employed/unemployed compared with being in school or in formal employment (table 1). In contrast, in women, HD and SU were more prevalent among those who were never married compared with those who had ever been married, in those who had a post-secondary school education and in those who were in school or formal employment compared with those who were unemployed/informally employed (table 2).

### Prevalence of risky sexual behaviour in people reporting HD and/or SU

Sexual behaviour of male and female HD and SU are shown in online supplemental table 1. Among men, a higher proportion of those who engaged in HD (50.9%), SU (43.8%) and concurrent HD and SU (61.1%) had had more than one sexual partner in the past 12 months compared with those who did not engage in HD or SU (21.4%). Similarly, a higher proportion of men who engaged in HD (6.6%), SU (5.2%) or concurrent SU and HD (13.5%) had received or given transactional sex compared with those who had neither reported HD or SU (1.9%) (online supplemental table 1). A similar trend was seen among female participants despite the small numbers of women who reported HD, SU and HD and SU.

While a lower proportion of sexually active men who engaged in HD and SU used condoms consistently, the opposite was observed among women, with a higher proportion of female HD or SU reporting consistent condom use. Women who engaged in HD (42.5%), SU (28.9%) and both HD and SU (42.8%) reported consistent condom use compared with non-HD and SU (23.2%).

**Table 2** substance and hazardous alcohol use of female participants against socio-demographic and economic characteristics of participants, by sex

| | Females | | | |
| | No SU or HD | SU only | HD only | HD and SU |
| Variable | N* | n* (%)† | n* (%)† | n* (%)† | n* (%)† |
|---|---|---|---|---|---|
| Total | 10701 | 10343 (97.0) (95% CI: 96.6 to 97.3) | 133 (1.12) (95% CI: 0.91 to 1.37) | 174 (1.49) (95% CI: 1.26 to 1.77) | 51 (0.41) (95% CI: 0.30 to 0.56) |
| Age category in years | | | | | |
| 18–20 | 5374 | 5213 (97.3) | 67 (1.2) | 66 (1.1) | 28 (0.4) |
| 21–24 | 5327 | 5130 (96.7) | 66 (1.0) | 108 (1.9) | 23 (0.4) |
| Marital status | | | | | |
| Never married | 6712 | 6448 (96.5) | 101 (1.3) | 124 (1.7) | 39 (0.5) |
| Ever married | 3989 | 3895 (97.7) | 32 (0.8) | 50 (1.2) | 12 (0.3) |
| Education level | | | | | |
| Up to primary school | 618 | 607 (98.3) | 3 (0.4) | 3 (0.5) | 5 (0.8) |
| Secondary school | 9317 | 9012 (97.0) | 118 (1.1) | 146 (1.4) | 41 (0.4) |
| Post-secondary schooling | 766 | 724 (95.0) | 12 (1.4) | 25 (3.1) | 5 (0.5) |
| Wealth quintile | | | | | |
| Poorest | 2499 | 2438 (97.9) | 21 (0.8) | 30 (1.0) | 10 (0.3) |
| Poorer | 2214 | 2150 (97.3) | 20 (0.9) | 37 (1.6) | 7 (0.3) |
| Middle | 2125 | 2065 (97.4) | 22 (0.9) | 30 (1.4) | 8 (0.3) |
| Richer | 1987 | 1914 (96.5) | 27 (1.3) | 36 (1.7) | 10 (0.5) |
| Richest | 1860 | 1763 (95.2) | 42 (2.1) | 40 (2.0) | 15 (0.7) |
| Employment | | | | | |
| In school or formal employment | 2934 | 2815 (96.5) | 49 (1.5) | 56 (1.7) | 14 (0.3) |
| Informal employment or unemployed | 7767 | 7528 (97.2) | 84 (1.0) | 118 (1.4) | 37 (0.4) |

*Unweighted count.
†Weighted per cent.
HD, hazardous drinking; SU, substance use.

## Association of hazardous drinking and substance use with sexual behaviour

Both men and women who engaged in HD and SU had significantly higher odds of having had more than one sexual partner in the past 12 months and engaging in transactional sex (tables 3 and 4). In men, after adjusting for socio-demographic factors, there were increased odds of having >1 sexual partner among HD (adjusted OR (aOR)=3.39, 95% CI: 2.70 to 4.26), SU (aOR=2.67, 95% CI: 2.21 to 3.23) and among those that did both (aOR=4.57, 95% CI: 3.59 to 5.81) compared with non-HD and SU. In addition, when comparing to men who did not engage in HD or SU, there was an increased odds of receiving/providing transactional sex among HD (aOR=3.62, 95% CI: 2.26 to 5.79), SU (aOR=2.49, 95% CI: 1.67 to 3.73) and concurrent HD and SU (aOR=7.72, 95% CI: 5.42 to 11.01).

In men but not women, SU was associated with a 22% increased odds of inconsistent condom use (aOR=1.22, 95% CI: 1.03 to 1.47). However, in women, HD only was associated with a 28% decreased odds of inconsistent condom use (aOR=0.72, 95% CI: 0.53 to 0.98).

## DISCUSSION

In this population-based survey of Zimbabwean youth, the prevalence of HD, SU and concurrent HD and SU was high, particularly in men who were older, in a higher wealth quintile and informally or unemployed. HD and SU were associated with having more than one sexual partner and having received or provided transactional sex in the past 12 months in both men and women. SU was associated with inconsistent condom use in men whereas HD in women was associated with consistent condom use.

A systematic review of population-based studies conducted between 2000 and 2016 in SSA reported a prevalence of lifetime use of any substance of 37% among adolescents in Southern Africa.[4] The considerable difference in SU prevalence between this and that reported in our study could be attributed to the inclusion of caffeine as a psychoactive substance in the systematic review, which is not usually the case in many studies analysing SU. Cannabis was the most commonly used substance in this study, results that are similar to other studies that have shown cannabis to be the most frequently used substance in Zimbabwe and in the Southern African

**Table 3** Association of substance use and hazardous alcohol drinking with sexual risk behaviours in male participants

| Variable | Sexual risk behaviours | | | | | |
| | More than one sexual partners in past 12-month (yes) (N*=6761) | | Transactional sex (yes) (N*=6880) | | Inconsistent condom use (yes) (N*=3729) | |
| | aOR† (95% CI) | P value | aOR‡ (95% CI) | P value | aOR§ (95% CI) | P value |
|---|---|---|---|---|---|---|
| Non-substance use and non-hazardous alcohol drinker | Reference | | Reference | | Reference | |
| Substance user only | 2.67 (2.21 to 3.22) | <0.001 | 2.51 (1.68 to 3.74) | <0.001 | 1.22 (1.03 to 1.47) | 0.024 |
| Hazardous alcohol drinker only | 3.40 (2.71 to 4.28) | <0.001 | 3.60 (2.24 to 5.79) | <0.001 | 0.84 (0.57 to 1.24) | 0.376 |
| Both hazardous alcohol drinker and substance user | 4.57 (3.59 to 5.81) | <0.001 | 7.74 (5.44 to 11.0) | <0.001 | 1.31 (0.95 to 1.82) | 0.100 |

*Unweighted count.
†OR adjusted for age group, marital status, education level, employment status, trial arm and province.
‡OR adjusted for age group, employment status, wealth quintile, trial arm and province.
§OR adjusted for age group, marital status, trial arm and employment status.
aOR, adjusted OR.

region.[24–26] When considering HD, results from our study were higher than those found from a study conducted in Eastern Africa in 2014, which showed a median prevalence of HD of 3% among young people 15–24 years of age.[27] In Zimbabwe in 2016, the population prevalence of alcohol use disorders in those 15 years and older was reported to be 6.4%.[24]

Consistent with our study findings, alcohol and SU has been reported to be more prevalent in men than women, with studies showing that current and ever use of substances can be more than three times higher in men than in women.[28–32] Increased HD and SU in men have been attributed to men having increased access and

opportunity to use alcohol and illicit SU, cultural roles and expectations surrounding masculinity and alcohol use or SU and the tendency for men to take part in risky health behaviours such as HD and SU.[29 31–33] It is important to note that the prevalence reported particularly among women, may be an underestimate due to the stigma attached to alcohol use and SU in this setting, which may have resulted in social desirability bias.[31 34]

We found a higher prevalence of HD and SU in men who were in a higher wealth quintile and those who were either informally employed or unemployed compared with being in school or having formal employment. Men have a breadwinning role and the higher prevalence of HD and

**Table 4** Association of substance use and hazardous alcohol drinking with sexual risk behaviours in female participants

| Variable | Sexual risk behaviours | | | | | |
| | More than one sexual partners in past 12-month (yes) (N*=10618) | | Transactional sex (yes) (N*=10701) | | Inconsistent condom use (yes) (N*=6529) | |
| | aOR† (95% CI) | P value | aOR‡ (95% CI) | P value | aOR§ (95% CI) | P value |
|---|---|---|---|---|---|---|
| Non-substance use and non-hazardous alcohol drinker | Reference | | Reference | | Reference | |
| Substance user only | 6.92 (4.39 to 10.9) | <0.001 | 6.62 (3.42 to 12.8) | <0.001 | 1.16 (0.67 to 2.03) | 0.589 |
| Hazardous alcohol drinker only | 5.61 (3.54 to 8.87) | <0.001 | 7.87 (4.48 to 13.8) | <0.001 | 0.72 (0.53 to 0.98) | 0.038 |
| Both hazardous alcohol drinker and substance user | 9.67 (4.37 to 21.4) | <0.001 | 14.3 (6.99 to 29.1) | <0.001 | 0.90 (0.45 to 1.79) | 0.766 |

*Unweighted count.
†OR adjusted for age group, marital status, employment status, education level, trial arm and province.
‡OR adjusted for age group, education level, employment status, trial arm and marital status.
§OR adjusted for age group, marital status, trial arm, education level, employment status, wealth quintile and province.
aOR, adjusted OR.

SU among young men may be a consequence of the lack of opportunities, financial instability and perceived hopelessness in a fragile economic environment with very high rates of unemployment particularly among young people.[35–38] In addition, increased supply and access of substances caused by porous borders and potentially the COVID-19 pandemic that resulted in youths being anxious, depressed or unoccupied because of being away from school or work due to the lockdowns or loss of employment, may have contributed to HD and SU in Zimbabwean youths.[35 37]

Interestingly, HD and SU although generally much less prevalent in women, were found to be more common in those who were in school or in formal employment. This may reflect women's ability to access these substances due to their ever-changing economic role in society that is brought about by education and formal employment, both of which give women the financial means to access differing substances more readily.[30]

HD and SU were associated with risky sexual behaviour, namely having multiple sexual partners and transactional sex. This finding is similar to that of other studies that found that heavy drinking and the use of cannabis and other illicit drugs are associated with an increased likelihood of having multiple sexual partners and having transactional sex in African youth.[39–43] This may reflect a group that is generally more 'risk-taking' and therefore more likely to engage in both SU and risky sexual behaviour at this age. HD and SU may also influence sexual risk behaviour most likely through their disinhibiting effect, resulting in the inability to reason and make decisions.[16 44]

Experimental alcohol use and SU as well as sexual behaviour tends to begin during adolescence and continue into young adulthood as young people gain autonomy and it is therefore a timely period to intervene to modify behaviours.[12 29] Education programmes, highlighting the dangers of HD and SU, need to start as early as possible, targeting both children and adolescents with age-appropriate messaging. This could be integrated with comprehensive sexuality education with a focus on young men, who are a particularly important target group given that they are generally more likely to engage in higher risk behaviours.

There is also a real scarcity of HD and SU treatment programmes in SSA, with many countries within the region lacking the capacity to meet the local demand for services.[45] Our study adds to the growing body of literature highlighting the growing problem of HD and SU in Africa, particularly among young people. Effective and context-appropriate interventions to address SU and HD are urgently needed. Interventions need to be affordable and accessible to all, despite gender differences, offering quality services, free from the risk of victimisation and related stigma.[28 45] Context-specific research on the range of substances used, the patterns and drivers of alcohol and SU and the context and culture of SU and HD among youth can inform interventions for both prevention and treatment of addiction.

Our study highlights the relationship between HD and SU and sexual behaviours and draws attention to both the need and opportunity for the integration of HD and SU screening in sexual reproductive healthcare services and as part of HIV prevention programmes. For example, when providing sexual health counselling, offering condoms, pre-exposure prophylaxis or HIV or STI testing, there is an opportunity to screen for and address SU.[46 47] Similarly, it is important to address the sexual health of individuals being treated for HD or SU.

Our study has a number of strengths, namely the large sample size and that the estimates were derived from a population-based survey. Participants were randomly selected, and participation rates were high, minimising selection bias. We acknowledge several limitations: due to its cross-sectional nature, we are unable to establish causality, or the temporality of associations observed. The use of self-reported measures may result in underreporting of alcohol and SU, as well as risky sexual behaviours. The study was conducted in urban and peri-urban settings and is therefore not representative of rural settings. In addition, we used a broad measure of SU, not taking into consideration the frequency or patterns of SU in analyses. Frequency and use of different substances (which may have different effects) were considered a single category.

## CONCLUSION

In conclusion, this study demonstrates an association between HD, SU and risky sexual behaviour. There is an urgent need for evidence-based, age, gender and context-appropriate preventive and treatment interventions targeting HD, SU and sexual behaviour. Importantly addressing SU and HD should be a core aspect of the HIV prevention toolkit and sexual health should be integrated within the management of SU and HD.

**Author affiliations**
[1]Biomedical Research and Training Institute, Harare, Zimbabwe
[2]University of Zimbabwe Faculty of Medicine and Health Sciences, Harare, Zimbabwe
[3]Department of Infectious Disease Epidemiology, London School of Hygiene & Tropical Medicine, London, UK
[4]Department of Public Health, Environments and Society, London School of Hygiene & Tropical Medicine, London, UK
[5]Clinical Research Department, London School of Hygiene & Tropical Medicine, London, UK

**Acknowledgements** The abstract from this paper was presented at INTEREST 2023 in Maputo Mozambique as poster number 113. The authors would also like to thank all the participants in the study.

**Contributors** KH conceptualised and is the lead investigator and author of this publication. SCA and VS analysed the data. RF is the Principal Investigator of the CHIEDZA Trial. and is the stduy guarantor. TB was responsible for data management. RF, CDC, ED, MT, CM and KK contributed to the coordination of the study and critical revisions of the manuscript. All authors read and approved the final manuscript.

**Funding** The CHIEDZA study was funded by the Wellcome Trust through a Senior Fellowship to RAF (206316/Z/17/Z).

**Competing interests** None declared.

**Patient and public involvement** Patients and/or the public were not involved in the design, or conduct, or reporting, or dissemination plans of this research.

**Patient consent for publication** Not applicable.

**Ethics approval** Ethical approval for the parent protocol was obtained from the Medical Research Council of Zimbabwe (reference number: MRCZ/A/2387), the Institutional Review Board of the Biomedical Research and Training Institute (reference number: AP149/2018) and the London School of Hygiene & Tropical Medicine (LSHTM) Research Ethics Committee (reference number: 12063). Participants viewed an information video about the study (in either English, Shona or Ndebele) on a tablet. Consent was documented electronically on a tablet, with participants retaining a signed paper copy for their records. Participants gave informed consent to participate in the study before taking part.

**Provenance and peer review** Not commissioned; externally peer reviewed.

**Data availability statement** Data are available upon reasonable request. The data sets used and/or analysed during the current study are available from the corresponding author on request.

**ORCID iDs**
Kudzai Hlahla http://orcid.org/0009-0006-9052-6488
Victoria Simms http://orcid.org/0000-0002-4897-458X
Chido Dziva Chikwari http://orcid.org/0000-0003-1617-3603
Mandikudza Tembo http://orcid.org/0000-0002-4520-3317

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
