## [Reviewer comments · BMJ Open]

This paper was submitted to a another journal from BMJ but declined for publication following peer review. The authors addressed the reviewers' comments and submitted the revised paper to BMJ Open. The paper was subsequently accepted for publication at BMJ Open.

ARTICLE DETAILS

TITLE (PROVISIONAL)	Prevalence of substance and hazardous alcohol use and their association with risky sexual behaviour among youth: findings from a population-based survey in Zimbabwe
AUTHORS	HLAHLA, KUDZAI; Azizi, Steven Chifundo; Simms, Victoria; Dziva Chikwari, Chido; Dauya, E; Bandason, Tsitsi; Tembo, Mandikudza; Mavodza, Constanca; Kranzer, Katharina; Ferrand, Rashida

VERSION 1 – REVIEW

REVIEWER	Nasui, Bogdana Adriana Iuliu Hațieganu University of Medicine and Pharmacy, Community Health
REVIEW RETURNED	17-Dec-2023

GENERAL COMMENTS	Dear Authors; Thank you for the opportunity to review this paper. The article is well-written, and the statistical tests are robust. However please find some comments: Page 6/30 Line 58 STIs is the abbreviation of what? Methods Page 8/30 is wealth quintile or quantile? Please explain how it was estimated (be more explicit). Also, give explanations about the stratification of demographic variables (not only in the table) Results Please insert in the body text both percentages and the number of respondents e.g. page 10/30 line 28, 29 ; page 12 line 8, line 13, line 16, etc. Table 2 Ever taken PrEp is the abbreviation of what?
---

REVIEWER	Marandure , Blessing N De Montfort University, School of Applied Social Sciences
REVIEW RETURNED	22-Jan-2024

GENERAL COMMENTS	Overall, the paper reports on an excellent piece of research conducted to a very high standard. There is a dearth of population-based data on substance use and hazardous drinking among Zimbabwean youth, hence this present paper plugs this gap (and goes some way towards developing a local evidence base for developing locally appropriate solutions). It is a shame that this has not been made apparent as highlighted in my comments
--

below, as without this prior knowledge other readers will not be able to ascertain this. Nevertheless, the paper is also timely, due to identified increases in substance use in Africa, and in particular Sub-Saharan Africa. Some specific points to address are provided below.

Introduction

-The introduction is far too brief, and doesn't provide a solid foundation for the present study. While the points raised in the second paragraph (lines 28-53) are pertinent, they are not sufficiently developed. For example, a brief exposition of the link between adolescence/young adulthood, cognitive development and substance use was needed.

-Lines 31 – 33 alludes to a potential link between adolescent risk taking/experimentation and substance use. Given the well-established literature making this link, together with related concepts such as sensation seeking, I would have expected a stronger statement being made here. This is a consequence of an insufficient review of pertinent literature highlighted above.

-The rationale for the present study needs further development. It would help to incorporate a brief review of previous literature linking substance use, heavy drinking and risky sexual behaviour. This will help to identify where the gaps are and how the current study adds to these. At present the novel addition to the literature this study adds is not made apparent.

Methods

-Overall, the methods section was well detailed, though could be brief in places. Detail not pertinent to the present analysis e.g. lines 13-15 referring to blood spot analysis could be removed. As the protocol has been published elsewhere, descriptions of sampling etc could be more concise.

-Line 33- source of/rationale for AUDIT cut-off score should be provided (ie. Was this based on the standard cut off points for the AUDIT? Specify and provide citation)

-Line 35- it would be beneficial to identify the substance use options that participants were presented with.

Results

-Lines 8-16 repeat information provided in the figure. This info can be deleted here for concision.

-Line 38-40 identifies weed/ganja as most prevalent substance. Prevalence rates for the other substances assessed for would aid in understanding patterns of substance use in the sample (and therefore which substances results most relate to).

Discussion

-p16 lines 48-55 identifies lower SU prevalence reported in the Zimbabwe Mental Health Investment Case, however does not offer explanations for the difference with that reported in the present study.

-p18 line 16 typo '...economic role of in society...

-p18 lines 10-18- it's not clear how women's 'ever changing role in society'... links to substance use in school/formal employments. This needs clarification.

-p19 line 3 identifies a scarcity in treatment programmes for HD and SU treatment programs. These needs contextualising. In what context has this scarcity been identified? Locally? Globally?

-the broad measure of substance use utilised needs to be acknowledged as a limitation, as it lacks sensitivity. Specifically, it treats one time, occasional, regular, and dependent users as a homogenous group.

VERSION 1 – AUTHOR RESPONSE

REVIEWER #1 COMMENTS- Dr. Bogdana Adriana Nasui, Iuliu Hațieganu University of Medicine and Pharmacy

COMMENT	RESPONSE
Introduction	
Page 6/30 Line 58 STIs is the abbreviation of what	Have defined the abbreviation on page 7/70 line 56
Methods	
Page 8/30 is wealth quintile or quantile? Please explain how it was estimated (be more explicit). Also, give explanations about the stratification of demographic variables (not only in the table)	It is wealth quintile because the wealth distribution is divided into fifths. Quintile is a type of quantile (equal-sized segments), but we have replaced the word quantile with quintile in the paper. The explanation about the stratification of sociodemographic variables and how the wealth quintiles were calculated have been provided in the Method section, page 11/70
Results	
Please insert in the body text both percentages and the number of respondents e.g. page 10/30 line 28, 29 ; page 12 line 8, line 13, line 16, etc.	This has been done
Table 2 Ever taken PrEp is the abbreviation of what?	PrEP is pre-exposure prophylaxis. Have defined PrEP with the asterisk at the bottom of the table

REVIEWER #2 COMMENTS- Dr. Blessing N Marandure , De Montfort University

COMMENT	RESPONSE
Introduction	
The introduction is far too brief, and doesn't provide a solid foundation for the present study. While the points raised in the second paragraph (lines 28-53) are pertinent, they are	Thank you for this comment. We have included - More information regarding brain and cognitive changes during adolescence/young adulthood that may

not sufficiently developed. For example, a brief exposition of the link between adolescence/young adulthood, cognitive development and substance use was needed.	result in increased experimentation, sensation seeking resulting in risk taking behaviour ie HD, SU and risky sexual behaviour
Lines 31 – 33 alludes to a potential link between adolescent risk taking/experimentation and substance use. Given the well-established literature making this link, together with related concepts such as sensation seeking, I would have expected a stronger statement being made here. This is a consequence of an insufficient review of pertinent literature highlighted above.	Thank you for this comment. We have made the statement stronger to show the link between adolescence, risk taking behaviour and potential substance use
The rationale for the present study needs further development. It would help to incorporate a brief review of previous literature linking substance use, heavy drinking and risky sexual behaviour. This will help to identify where the gaps are and how the current study adds to these. At present the novel addition to the literature this study adds is not made apparent.	Thank you for this comment.  -We have included literature on current studies that have been conducted on this topic -We have highlighted the current gap that exists specifically in the Zimbabwean context and therefore the gap that the current study will fill
Methods	
Overall, the methods section was well-detailed, though could be brief in places. Detail not pertinent to the present analysis e.g. lines 13-15 referring to blood spot analysis could be removed. As the protocol has been published elsewhere, descriptions of sampling etc could be more concise.	The description of the survey sampling has been summarized as indicated on pages 9-12. We have removed the blood spot and any aspects that are not part of the present study.
-Line 33- source of/rationale for AUDIT cut-off score should be provided (ie. Was this based on the standard cut off points for the AUDIT? Specify and provide citation)	A citation has been provided on page 10, with 22 as the reference. An AUDIT score cut off of 8, according to WHO standards is indicative of HD or harmful alcohol use
-Line 35- it would be beneficial to identify the substance use options that participants were presented with.	The names of the substances have already been mentioned in lines 3-13 on page 10.
Results	
Lines 8-16 repeat information provided in the figure. This info can be deleted here for concision.	Thank you for the suggestion. We have decided to leave the information in as it may be useful for interpretation of the study and the figure.
-Line 38-40 identifies weed/ganja as most prevalent substance. Prevalence rates for the other substances assessed for would aid in understanding patterns of substance use in the	Agreed. Have added the prevalences of the top three substances used by both males and females

sample (and therefore which substances results most relate to).	
Discussion	
-p16 lines 48-55 identifies lower SU prevalence reported in the Zimbabwe Mental Health Investment Case, however does not offer explanations for the difference with that reported in the present study.	Thank you for this. We have now removed this reference as the information could not be verified
p18 line 16 typo '...economic role of in society...	Corrected. Thank you
p18 lines 10-18- it's not clear how women's 'ever changing role in society'... links to substance use in school/formal employments. This needs clarification.	Have expanded on how education and formal employment give women financial means to access alcohol and substances
p19 line 3 identifies a scarcity in treatment programmes for HD and SU treatment programs. These needs contextualising. In what context has this scarcity been identified? Locally? Globally?	I have added the context ie SSA and the inability of countries to provide SU treatment services
the broad measure of substance use utilised needs to be acknowledged as a limitation, as it lacks sensitivity. Specifically, it treats one time, occasional, regular, and dependent users as a homogenous group.	Very true. I have added this to the limitations

VERSION 2 – REVIEW

REVIEWER	Marandure , Blessing N De Montfort University, School of Applied Social Sciences
REVIEW RETURNED	14-May-2024

GENERAL COMMENTS	Dear Authors Thank you for a thorough and constructive response to reviewer comments. All issues have been addressed to a good standard.
---